

# Half-hourly changes in intertidal temperature at nine wave-exposed locations along the Atlantic Canadian coast: a 5.5-year study

Ricardo A. Scrosati, Julius A. Ellrich, Matthew J. Freeman

Department of Biology, St. Francis Xavier University, Antigonish, Nova Scotia B2G 2W5, Canada

*Correspondence to:* Ricardo A. Scrosati (rscrosat@stfx.ca)

**Abstract.** Intertidal habitats are unique because they spend alternating periods of submergence (at high tide) and emergence (at low tide) every day. Thus, intertidal temperature is mainly driven by sea surface temperature (SST) during high tides and by air temperature during low tides. Because of that, the switch from high to low tides and viceversa can determine rapid changes in intertidal thermal conditions. On cold-temperate shores, which are characterized by cold winters and warm summers, intertidal thermal conditions can also change considerably with seasons. Despite this uniqueness, knowledge on intertidal temperature dynamics is more limited than for open seas. This is especially true for wave-exposed intertidal habitats, which, in addition to the unique properties described above, are also characterized by wave splash being able to moderate intertidal thermal extremes during low tides. To address this knowledge gap, we measured temperature every half hour during a period of 5.5 years (2014-2019) at nine wave-exposed rocky intertidal locations along the Atlantic coast of Nova Scotia, Canada. This data set is freely available from the figshare online repository (Scrosati and Ellrich, 2020a; https://doi.org/10.6084/m9.figshare.12462065.v1). We summarize the main properties of this data set by focusing on location-wise values of daily maximum and minimum temperature and daily SST, which we make freely available as a separate data set in figshare (Scrosati et al., 2020; https://doi.org/10.6084/m9.figshare.12453374.v1). Overall, this cold-temperate coast exhibited a wide annual SST range, from a lowest overall value of -1.8 °C in winter to a highest overall value of 22.8 °C in summer. In addition, the latitudinal SST trend along this coast experienced a reversal from winter, when SST increased southwards, to summer, when SST decreased southwards, seemingly driven by alongshore differences in coastal upwelling. Daily temperature maxima and minima were more extreme, as expected from their occurrence during low tides, ranging from a lowest overall value of -16.3 °C in winter to a highest overall value of 41.2 °C in summer. Daily maximum temperature in summer varied little along the coast, while daily minimum temperature in winter increased southwards. This data set is the first of its kind for the Atlantic Canadian coast and exemplifies in detail how intertidal temperature varies in wave-exposed environments on a cold-temperate coast.

## 1 Introduction

Rocky intertidal habitats occur on marine rocky shores between the highest and lowest elevations reached by tides. These environments are unique because they spend alternating periods of submergence (during high tides) and emergence (during low tides) every day (Raffaelli and Hawkins, 1999; Menge and Branch, 2001). Thus, on the one hand, intertidal conditions are influenced by the seasonal changes in sea surface temperature (SST), which can be pronounced on temperate shores, which display warm waters in summer but cold waters in winter. On the other hand, an even greater degree of thermal variation can occur at hourly scales



once intertidal habitats become exposed to the air at low tide, especially on hot days in spring and summer (Watt and Scrosati, 2013; Lathlean et al., 2014; Umanzor et al., 2017) and cold days in winter (Scrosati and Ellrich, 2018a).

Temperature is a major factor influencing the distribution and abundance of species (Pörtner, 2002; Körner et al., 2016; Lancaster and Humphreys, 2020). Thus, SST plays an important ecological role in intertidal habitats during high tides (Sanford, 2014), while high (Somero 2007) and low (Braby 2007) air temperatures are ecologically relevant during low tides. In addition, not only average temperature matters, but its temporal variability as well (Bennedetti-Cecchi et al., 2006). Overall, then, having detailed temperature data across periods of low and high tide is important for intertidal ecology and to make biogeographic predictions based on climate change expectations (Wethey et al., 2011).

Temperature data are available for surface ocean waters worldwide (Fay and McKinley, 2014; Banzon et al., 2016; Freeman and Lovenduski, 2016; Aulicino et al., 2018; Yun et al., 2019). However, data on intertidal temperature are considerably less common, both in terms of spatial and temporal coverage (Lathlean et al., 2014; Umanzor et al., 2017; Scrosati and Ellrich, 2018a). This is especially true for wave-exposed intertidal habitats, as remote sensing methods that are commonly used for open waters (e.g., satellites) cannot capture the quick, localized temperature changes caused by tides and waves on exposed shores. Waves can also damage equipment deployed in-situ to measure intertidal temperature. For wave-exposed intertidal habitats, temperature data between consecutive low and high tides can also be used to infer physical aspects of the environment such as wave action itself (Harley and Helmuth, 2003).

Wave-exposed rocky intertidal habitats are common along the Atlantic Canadian coast in Nova Scotia, as this coast faces the open ocean. Several studies have investigated the ecology of these environments (Minchinton and Scheibling, 1991; Hunt and Scheibling, 1998, 2001; Scrosati and Heaven, 2007; Arribas et al., 2014; Molis et al., 2015; Ellrich and Scrosati, 2016; Scrosati and Ellrich, 2018b, 2019; Scrosati, 2020). However, because of their research goals, intertidal temperature was either not measured or analyzed for a few locations or limited time periods. Therefore, there is a knowledge gap on broad spatio-temporal patterns in intertidal temperature for wave-exposed environments along this coast. To address this gap, in this article we provide and discuss a data set consisting of intertidal temperature values measured every half hour at nine wave-exposed locations along the Atlantic coast of Nova Scotia spanning a period of 5.5 years.

## 2 Methods

We monitored intertidal temperature at nine locations that span the full extent of the open Atlantic coast of mainland Nova Scotia, nearly 415 km (Fig. 1). For simplicity, these locations will hereafter be referred to as L1 to L9, from north to south. Their names and coordinates are provided in Table 1. The substrate of these intertidal locations is stable bedrock. All of them face the open Atlantic Ocean without physical obstructions, so they are wave-exposed. Values of daily maximum water velocity (an indication of wave exposure) measured with dynamometers (see design in Bell and Denny, 1994) in wave-exposed intertidal habitats from this coast range between 6-12 m s$^{-1}$ (Hunt and Scheibling, 2001; Scrosati and Heaven, 2007; Ellrich and Scrosati, 2017).



We started to monitor intertidal temperature in April-May 2014 at L2-L9 and in April 2015
at L1 (see the precise dates in Scrosati and Ellrich, 2020a). We measured temperature with
submersible loggers (HOBO Pendant logger, Onset Computer, Bourne, MA, USA) that were
kept attached to the intertidal substrate with plastic cable ties secured to eye screws drilled into
the substrate, allowing almost no contact between the loggers and the substrate. We kept the
substrate around the loggers always free of macroalgal canopies and sessile invertebrates. To
have a continuous temperature record during the 5.5 years of this study, we replaced the loggers
periodically. At each location, we installed replicate loggers several meters apart from one
another at an elevation just above the mid-intertidal zone. We set the loggers to record
temperature every 30 min. We stopped recording temperature in November 2018 at L1 and L3
and in August-October 2019 at L2 and L4-L9 (see the precise dates in Scrosati and Ellrich,
2020a). For each location, temperature was highly correlated between the replicate loggers
during the study period (mean $r = 0.97$). Thus, we averaged the corresponding half-hourly
values to generate one time series of half-hourly temperature data for each location for the
studied period, which is the data set on which this paper is based (Scrosati and Ellrich, 2020a).
Due to its high temporal resolution, this data set could be used in the future for a variety of
purposes. To summarize its main properties here, we extracted values that are commonly used in
intertidal ecology and coastal oceanography and that therefore could be of immediate interest:
daily maximum and minimum temperature (MaxT and MinT, respectively) and daily SST. As
the Nova Scotia coast is cold-temperate, we expected SST to be often considerably lower than
MaxT in spring and summer, as MaxT is then reached during low tides when intertidal
environments are usually exposed to high air temperatures. Conversely, we expected SST and
MaxT to be more similar or even the same in winter, as low tides then often expose intertidal
habitats to negative air temperatures below the freezing point of seawater. For these same
reasons, we also expected SST to be typically higher than MinT in winter, as MinT is then
generally reached during low tides, but more similar to MinT in spring and summer. For each
location, we extracted the values of daily MaxT, MinT, and SST from the corresponding set of
half-hourly data on intertidal temperature (Scrosati and Ellrich, 2020a). We considered daily
SST as the temperature recorded closest to the time of the highest tide of each day, as the
loggers were then fully submerged in seawater. We determined the time of such tides using
information (Tide and Current Predictor, 2020) for the tide reference stations that are closest to
our intertidal locations (Table 1).
**3 Main patterns in the data and relevance to future research**
We obtained half-hourly temperature data during the monitoring period specified above for
each location with just two exceptions: the period between 20 March and 12 April 2017 for L1
because of logger removal by drift sea ice coming from the Gulf of St. Lawrence and the period
between 30 September 2014 and 26 April 2015 for L9 because of logger loss caused by wave
action. Continued monitoring after both such periods was possible after installing new loggers.
This data set is available online (Scrosati and Ellrich, 2020a).
The temporal changes in daily MaxT, MinT, and SST during the studied period at each
location are shown in Fig. 2. For convenience, all of these daily values are also available from
the figshare online repository (Scrosati et al., 2020). The highest and lowest values of SST for
each location (Table 2) reveal that this cold-temperate coast has a wide seasonal range of SST
(see worldwide SST ranges in figure 6.3 in Stewart, 2008). The highest location-wise values of



SST occurred in summer and ranged between 20 °C and 22.8 °C, while the lowest location-wise
SST values occurred in winter and were near the freezing point of seawater, between -0.9 °C
and -1.8 °C (Table 2, Fig. 2). We note that, unlike the nearby Gulf of St. Lawrence (Fig. 1;
Saucier et al., 2003) or wave-sheltered coves along the Atlantic coast of Nova Scotia, open
waters washing wave-exposed habitats along the Atlantic coast of Nova Scotia do not freeze in
winter (Canadian Ice Service, 2020). Overall, for the studied period, the location-wise difference
between the highest and lowest SST values ranged between 21.1 °C and 24.6 °C. Although there
was some patchiness in this seasonal SST range along the coast, it was lowest in two southern
locations (L7 and L9) driven by lower values of maximum summer SST there (Table 2).
The occurrence of the lowest location-wise values of maximum summer SST at two
southern locations (L7 and L9) is related to a broader alongshore pattern. Based on the data for
the summer months (for convenience, July, August, and September) for the years when SST was
measured at all locations in those months (2015, 2016, 2017, and 2018), mean location-wise
SST in summer decreased from north to south, from 17.5 °C at L1 to 13.2 °C at L9 (Table 2). In
contrast, an equivalent analysis done for winter months (for convenience, January, February,
and March) for the years when SST was measured at all locations in those months (2016, 2017,
and 2018) revealed that mean location-wise SST in winter actually increased from north to
south, from 0.8 °C at L1 to 3.0 °C at L9 (Table 2). In other words, a summer-to-winter reversal
in the latitudinal trend in intertidal SST takes place on this coast, as waters are warmer in
summer and colder in winter in northern locations than in southern locations.
The southward decrease of intertidal SST in summer is likely influenced by alongshore
differences in coastal upwelling. On the Atlantic coast of Nova Scotia, upwelling-favourable
winds are more common in summer than in winter (Garrett and Loucks, 1976; Dever et al.,
2018). Although possible alongshore differences in upwelling have not been studied in detail,
they seem to exist. For example, Petrie et al. (1987) reported that seawater temperature at 6-20
m of depth decreased from June to July 1984 near L6–L7 because of wind-driven upwelling,
while temperature at those depths increased north of that coastal range during that period. More
recently, Shan et al. (2016) have also referred to wind-driven upwelling on the southeastern
Nova Scotia coast. A detailed analysis of daily changes in intertidal SST is beyond the
objectives of this data paper. However, Fig. 2 reveals basic differences in summer cooling
between northern and southern locations. Summer cooling events were generally marked in
southern locations, especially at L6 and L7, where SST could drop by 10 °C in 5-10 days, in
some cases reaching values below 5 °C (Fig. 2). An analysis of coastal winds at L6 and L7
indicated that wind-driven upwelling explained the cooling observed at those locations in July
2014 (Scrosati and Ellrich, 2020b). Although persistent, the summer cooling signal that was
often pronounced at L6 and L7 (Fig. 2) weakened progressively towards northern locations,
especially at L1 and L2. In fact, at L1, SST never dropped below 10 °C in summer months (Fig.
2). These considerations could orient future research to unravel the drivers of the latitudinal
changes in summer SST revealed by this study.
The southward increase of intertidal SST observed in winter could be a result of latitudinal
changes in heat flux from the atmosphere (Stewart, 2008; Deser et al., 2010; Shan et al., 2016),
although other processes are also generally at play in coastal environments (Hebert et al., 2016;
Larouche and Galbraith, 2016). For example, for the studied coast, the abundant sea ice formed
across the Gulf of St. Lawrence (Fig. 1) every winter (Saucier et al., 2003) may contribute to
keep intertidal SST low at our northern locations, as the waters that leave this gulf flow





southwards following the coast of mainland Nova Scotia (Han et al., 1997; Hebert et al., 2016;
Dever et al., 2018), reaching our northern locations first before they warm up on their way
south.
As expected from the warm summers and cold winters that characterize eastern Canada
(Government of Canada, 2020), MaxT was often considerably higher than SST in spring and
summer and MinT was often lower than SST in fall and winter (Fig. 2), as MaxT and MinT
typically take place at low tide during those respective seasons. The highest location-wise values
of MaxT almost doubled those of SST, as they ranged between 36.1°C and 41.2 °C. The lowest
location-wise values of MinT ranged between -9.1 °C and -16.3 °C. Therefore, the location-wise
difference between the highest and lowest daily temperatures, which ranged between 46.1 °C
and 54.4 °C, generally more than doubled the location-wise difference between the highest and
lowest daily SST values (Table 2).
The highest value of MaxT differed little among locations (Table 2). Based on the data for
the summer months (for convenience, July, August, and September) for the years when SST was
measured at all locations in those months (2015, 2016, 2017, and 2018), mean location-wise
MaxT in summer exhibited patchiness along the coast without any clear latitudinal trend (Table
2). As MaxT in summer generally occurs during aerial exposure at low tides, both climatic and
oceanographic influences may interact to determine its alongshore pattern. For instance, summer
values of MaxT might simply be expected to increase southwards following warmer air
temperatures on land (Government of Canada, 2020). However, the SST drops due to coastal
upwelling in southern locations in summer might actually temper air temperatures right on the
coast, thus limiting MaxT. In the end, climate and oceanography might together be responsible
for the patchy alongshore MaxT pattern, which seems dependent on local conditions.
Researching these possibilities could thus be of interest. In contrast, the data for winter months
(for convenience, January, February, and March) for the years when MinT was measured at all
locations in those months (2016, 2017, and 2018) revealed that mean location-wise MinT in
winter generally increased from north to south, the lowest such average (-2.7 °C) registered at
L1 and the highest one (0.2 °C) at L9 (Table 2). Thus, the alongshore pattern of winter MinT
may more clearly respond to typical latitudinal changes in winter air temperatures and perhaps
also to influences of Gulf of St. Lawrence sea ice (see above) on northern locations.
Another salient property of our data is that the daily changes in MaxT in spring and
summer and MinT in fall and winter were much larger than the corresponding daily changes in
SST (Fig. 2). Such a high day-to-day variability in MaxT and MinT likely reflects daily changes
in weather conditions, which affect intertidal habitats at low tides, as well as wave exposure, as
wave-generated splash during low tides on wavy days keep intertidal habitats wet and, thus,
often cooler than the air in summer and warmer than the air in winter. Thus, the interaction
between weather and wave action as a determinant of intertidal thermal extremes is another
research area deserving attention in the future. Ultimately, given the prominent role of extreme
abiotic events in ecology (Denny et al., 2009; Smith, 2011; Nowicki et al., 2019), the marked
daily changes in MaxT and MinT during those seasons highlight the potentially critical role of
low tides for the survival of intertidal organisms on these environmentally variable habitats.
Another interesting characteristic of our data set is that the daily average between MaxT
and MinT was generally higher than SST in spring and summer but generally lower than SST in
fall and winter (Fig. 2). In other words, the average intertidal temperature measured during low
tides increased faster from winter to summer and decreased faster from summer to winter than



SST. This difference likely reflects the difference in heat capacity between air and water, which
makes SST follow air temperatures throughout seasons with a delay (Stewart, 2008).
Our data set could also be useful to investigate climatic drivers of interannual differences in
intertidal temperature. For example, a marked difference in upwelling-driven coastal cooling at
L6 and L7 between July 2014 (strong) and July 2015 (weak) was related to a normal (2014)
versus El Niño (2015) conditions (Scrosati and Ellrich, 2020b). Although El Niño (ENSO) is
predominantly a Pacific phenomenon (Timmermann et al., 2018), it is also related to interannual
weather changes in North America through climatic teleconnections (George and Wolfe 2009;
Wu and Lin 2012; Whan and Zwiers 2017; Dai and Tan, 2019). Another large-scale climate
phenomenon, the North Atlantic Oscillation (NAO), influences weather patterns mainly in the
North Atlantic basin (Hanna and Cropper, 2017). It would thus be interesting to study whether
NAO and ENSO might interact (Wu and Lin, 2012; Nalley et al., 2019) to affect winds,
upwelling, and ultimately intertidal temperature along the Nova Scotia coast.
**4 Conclusions**
This is a unique data set because it describes intertidal temperature with a high temporal
resolution during a period of 5.5 years at nine wave-exposed locations spanning the full extent
of the Atlantic coast of mainland Nova Scotia. The main patterns described above have revealed
previously unknown latitudinal and seasonal trends in intertidal temperature on this coast. The
considerations on the possible mechanisms underlying these patterns should help orient future
research on the drivers of thermal variation in these intertidal environments. Because of the
temporal and spatial scales of this data set, we believe that future research using these data could
lead to theoretical advances in coastal oceanography and intertidal thermal ecology. Ultimately,
this data set represents a detailed baseline on which to study the influence of climatic and
oceanographic change on intertidal temperature variation in this cold-temperate system.
**Data availability**
The full data set on half-hourly temperature measured at the nine intertidal locations
between 2014 and 2019 is available from the figshare online repository (Scrosati and Ellrich,
2020a; https://doi.org/10.6084/m9.figshare.12462065.v1). The daily values of MaxT, MinT, and
SST for these locations during this time period are also available from the figshare online
repository (Scrosati et al., 2020; https://doi.org/10.6084/m9.figshare.12453374.v1).
**Author contributions**
RAS designed the study. RAS and JAE led field work and JAE and MJF data curation.
RAS wrote the manuscript and JAE and MJF reviewed it before submission.
**Competing interests**
The authors declare that they have no conflict of interest.
**Acknowledgements**
We thank Alexis Catalán, Carmen Denfeld, Willy Petzold, and Maike Willers for field
assistance.

## Financial support

This study was funded by grants awarded to RAS by the Natural Sciences and Engineering Research Council of Canada (NSERC Discovery Grant #311624), the Canada Research Chairs program (CRC grant #210283), and the Canada Foundation for Innovation (CFI Leaders Opportunity Grant #202034) and by a postdoctoral fellowship awarded to JAE by the German Academic Exchange Service (DAAD fellowship #91617093).

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

416

| Location code | Name of studied intertidal location (geographic coordinates) | Closest tide reference station (geographic coordinates) |
| --- | --- | --- |
| L1 | Glasgow Head (45.3203° N, 60.9592° W) | Canso (45.3500° N, 61.0000° W) |
| L2 | Deming Island (45.2121° N, 61.1738° W) | Whitehead (45.2333° N, 61.1833° W) |
| L3 | Tor Bay Provincial Park (45.1823° N, 61.3553° W) | Larry's River (45.2167° N, 61.3833° W) |
| L4 | Barachois Head (45.0890° N, 61.6933° W) | Port Bickerton (45.1000° N, 61.7333° W) |
| L5 | Sober Island (44.8223° N, 62.4573° W) | Port Bickerton (45.1000° N, 61.7333° W) |
| L6 | Duck Reef (44.4913° N, 63.5270° W) | Sambro (44.4833° N, 63.6000° W) |
| L7 | Western Head (43.9896° N, 64.6607° W) | Liverpool (44.0500° N, 64.7167° W) |
| L8 | West Point (43.6533° N, 65.1309° W) | Lockport (43.7000° N, 65.1167° W) |
| L9 | Baccaro Point (43.4496° N, 65.4697° W) | Ingomar (43.5667° N, 65.3333° W) |

417



**Table 2.** Summary values of daily MaxT, MinT, and SST (°C) for the nine wave-exposed
intertidal locations (L1 to L9, from north to south) surveyed between 2014 and 2019 along the
Atlantic Canadian coast (see Methods for details on how each row of values was determined).

| | L1 | L2 | L3 | L4 | L5 | L6 | L7 | L8 | L9 |
|---|---|---|---|---|---|---|---|---|---|
| Highest daily MaxT | 38.1 | 38.3 | 37.9 | 36.1 | 41.2 | 36.5 | 37.1 | 37.2 | 38.5 |
| Lowest daily MinT | -16.3 | -10.8 | -11.0 | -10.0 | -12.2 | -15.5 | -13.0 | -9.1 | -11.6 |
| Highest temperature range | 54.4 | 49.1 | 48.9 | 46.1 | 53.4 | 52.0 | 50.1 | 46.3 | 50.1 |
| Summer mean MaxT | 25.1 | 22.9 | 22.6 | 20.7 | 25.5 | 23.5 | 22.8 | 23.2 | 21.8 |
| Winter mean MinT | -2.7 | -1.4 | -1.3 | -0.4 | -2.2 | -1.2 | -0.3 | 0.02 | 0.2 |
| Highest daily SST | 22.5 | 21.5 | 21.8 | 22.8 | 22.2 | 21.9 | 20.3 | 22.1 | 20.0 |
| Lowest daily SST | -1.7 | -1.7 | -1.4 | -1.8 | -1.8 | -1.8 | -0.9 | -1.7 | -1.7 |
| Highest SST range | 24.2 | 23.2 | 23.2 | 24.6 | 24.0 | 23.7 | 21.1 | 23.7 | 21.7 |
| Summer mean SST | 17.5 | 16.4 | 16.1 | 16.1 | 15.8 | 15.2 | 13.3 | 14.2 | 13.2 |
| Winter mean SST | 0.8 | 1.0 | 1.3 | 1.3 | 1.6 | 2.2 | 2.7 | 2.8 | 3.0 |


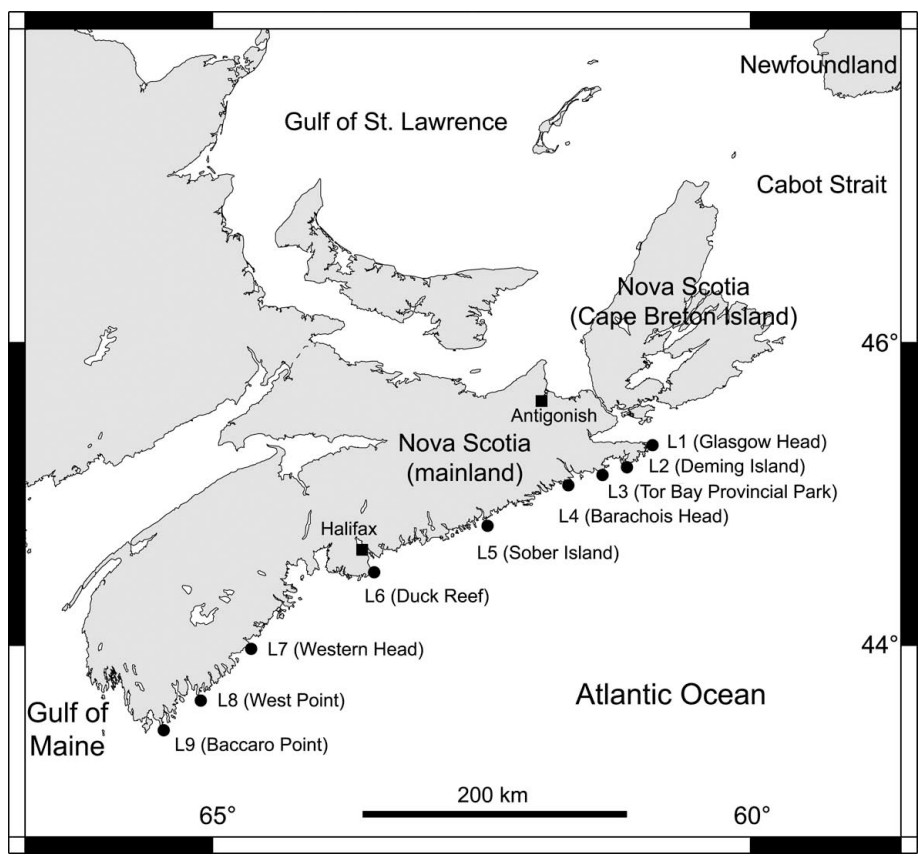

**Figure 1.** Map indicating the position of the nine wave-exposed intertidal locations surveyed along the Atlantic coast of mainland Nova Scotia, Canada.

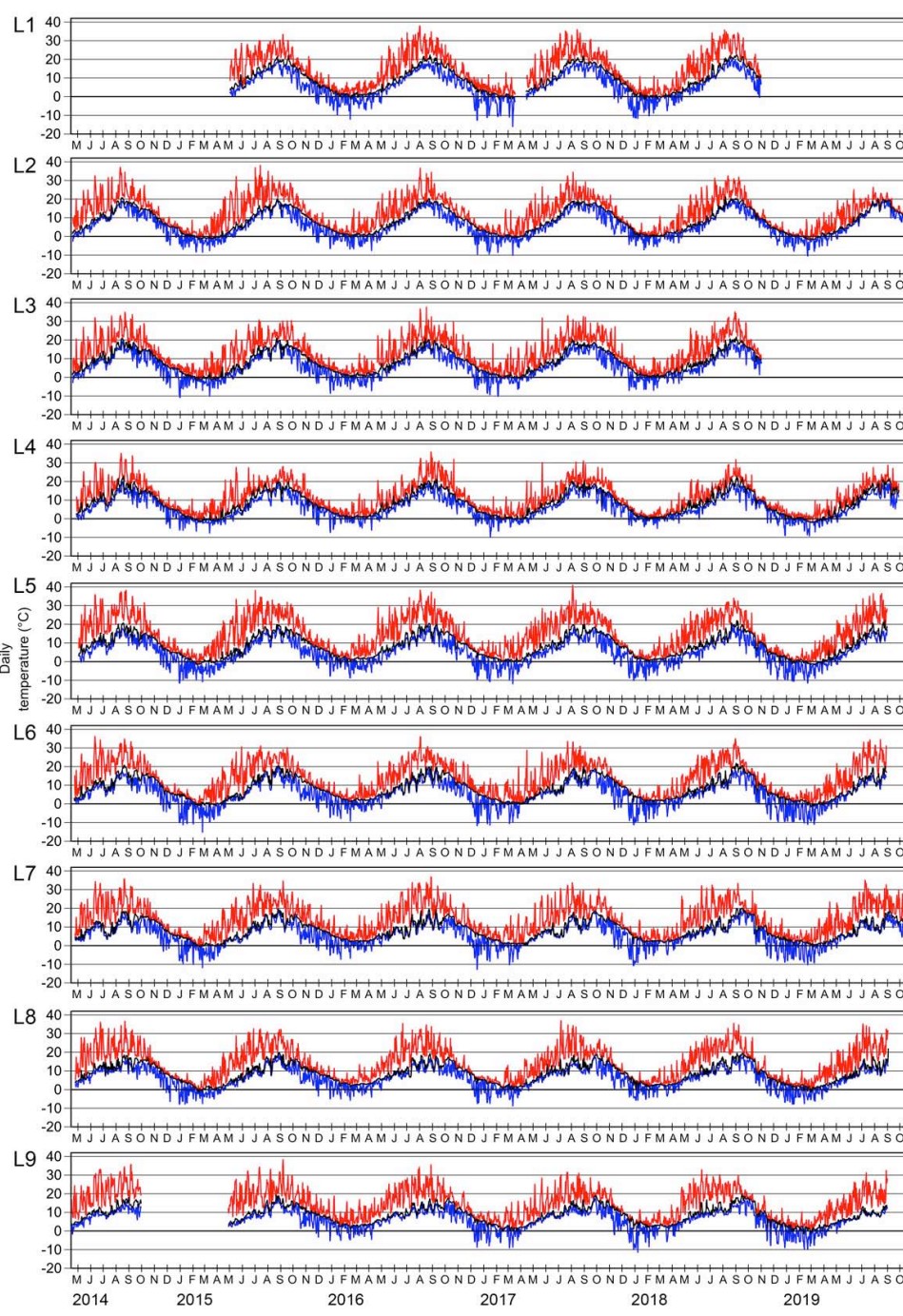

**Figure 2.** Daily MaxT (red line), MinT (blue line), and SST (black line) at the nine intertidal
locations (L1 to L9, from north to south) surveyed between April 2014 and October 2019.