# Peer review of "Half-hourly changes in intertidal temperature at nine wave-exposed locations along the Atlantic Canadian coast: a 5.5-year study"

_Earth System Science Data, 2020_

## Referee Comment (RC1) · Anonymous Referee #1 · 14 Jul 2020

Dear Editor Giuseppe M.R. Manzella, I am pleased to send the revision of the manuscript entitled "Half-hourly changes in intertidal temperature at nine wave-exposed locations along the Atlantic Canadian coast: a 5.5-year study" by Scrosati, Ellrich, and Freeman. The work is described with clarity and easily understandable. I found the results and conclusions relatively straight-forward and the manuscript as a whole in good shape.

As the authors highlight, there is a great lack of this type of information in intertidal rocky areas. This long data set on spatial and temporal patterns in intertidals can be useful in a climate change scenario to predict community changes and generate advances in

coastal oceanography. I therefore recommend publication following a minor revision.

Minor revisions: L 47: Add comma after author. L 82: I would like the authors incorporate some information of mean tidal amplitude. L 138-139 and L141-143: Detail of summer and winter month and years sampled must be moved to Methods. L 146: Authors could incorporate information of cold and warm near shore currents in the study area to highlight the results showed. L 185: Delete "for convenience, July, August, and September" L 196: Delete "for convenience, January, February, and March" Figure 2: Increase y-axis title size

---

## Author Comment (AC1) · 14 Aug 2020

Analysis of the review

Dear Editor Giuseppe M.R. Manzella, I am pleased to send the revision of the manuscript entitled "Half-hourly changes in intertidal temperature at nine wave-exposed locations along the Atlantic Canadian coast: a 5.5-year study" by Scrosati, Ellrich, and Freeman. The work is described with clarity and easily understandable. I found the results and conclusions relatively straight-forward and the manuscript as a whole in good shape. As the authors highlight, there is a great lack of this type of information in intertidal rocky areas. This long data set on spatial and temporal patterns

in intertidals can be useful in a climate change scenario to predict community changes and generate advances in coastal oceanography. I therefore recommend publication following a minor revision.

We are grateful for the reviewer's overall support. The specific comments are addressed below and the applied changes are highlighted in yellow in the revised manuscript.

—- Minor revisions: L 47: Add comma after author.

Done.

—- L 82: I would like the authors incorporate some information of mean tidal amplitude.

We have added detailed information on this in lines 92-108.

—- L 138-139 and L141-143: Detail of summer and winter month and years sampled must be moved to Methods.

We believe that those brief details make sense the most when describing those results, which explains why they are included in section 3, "Main patterns in the data".

—- L 146: Authors could incorporate information of cold and warm nearshore currents in the study area to highlight the results showed.

We have added information about the single current that washes the studied coast in lines 82-84.

—- L 185: Delete "for convenience, July, August, and September" L 196: Delete "for convenience, January, February, and March"

We believe that making those statements is important to determine unequivocally what specific months were used for those calculations, so readers can replicate the analyses.

—- Figure 2: Increase y-axis title size.

Done.

We appreciate the constructive comments from this reviewer, which we are stating in the revised Acknowledgements section.

---

## Referee Comment (RC2) · Anonymous Referee #2 · 16 Sep 2020

Summary: This manuscript presents a 5-year record of field observations for temperature across various rocky intertidal zones along the Atlantic Canadian coast. The purpose of the dataset is to address data and knowledge gaps in temperature fluctuations in intertidal regions that are influenced by seasonal factors, daily tidal oscillations, as well as large and small scale atmospheric processes. In this way, the dataset can be used to help advance our understanding of the processes affecting the ecology rocky intertidal zones, and how these processes are influence by broader climate change.

Overall Comments: The manuscript describing this dataset is very well written. The authors do a very nice job of explaining why the data are important and relevant and

how the field data were collected. I also believe the authors do a good job of interpreting their dataset to the audience. In particular, I like that the data are presented in terms of temperature variability across a range of spatial and temporal scales. For instance, for the temporal dimension, the authors use the data to show the general variation in temperature due to seasonal Earth-Sun relationships (i.e. peak temps occurring slightly after the summer solstice), and the hourly variations arising from the push and pull of the tides. Spatial variability is also highlighted by the data with regional variations due to upwelling and the small-scale effects of local weather. Furthermore, the authors use the data to highlight the broader effects of global atmospheric circulation processes (i.e. ENSO; NAO) on rocky intertidal zone temps. This adds broader impacts of the dataset in that it can help move forward modeling studies to address, for instance, the effects of climate change on these ecosystems.

I do not know of any other dataset quite like this one, so I do believe it is new. However, the methods are not necessarily novel. But, that is OK I think. The quality of the dataset are good and the links are accessible. The authors use replicate loggers and run statistics across the datasets with high correlation between sites over the study period. Figures and table look good and are appropriate.

Rating: Uniqueness: 1 – Again, I do not know of any other dataset like this one, and I do believe that it "provides data on a variable that is supposed to reflect changes in the Earth system". Usefulness: 1 – I believe that this data could be used alone to compare to future trends to assess changes in the Earth system, and can also be combined with other datasets (such as those from a coastal zone observatory) to develop numerical models to predict such changes and assess the forcing mechanisms behind that change. Completeness: 1 – data are complete and do not appear to be split across multiple manuscripts.

Minor (technical) considerations: L48-49: The wording in this sentence is a bit awkward. Consider rephrasing? L97: "which is the data set on which this paper is based" → I'm not sure you need to say this here? Maybe delete this statement for better flow?

[Figure]

L117 – 121: consider adding numbers to the two exceptions. It was a little hard to dig these out of the text and I think adding a number between each will help pull it out and make it clearer for the reader. For instance, "…with just two exceptions: 1) the period between 20 March and 12 April 2017 for L1 because of logger removal by drift sea ice coming from the Gulf of S. Lawrence, and 2) the period between 30 September…."

―――――――――――――――――――――

---

## Author Comment (AC3) · 20 Sep 2020

Analysis of the comments by reviewer 2

The changes that we have made in our manuscript to address the reviewer's comments are discussed below and are highlighted in the revised manuscript IN GREEN COLOUR.

—

COMMENT FROM REVIEWER: Summary: This manuscript presents a 5-year record of field observations for temperature across various rocky intertidal zones along the

[Figure]

Atlantic Canadian coast. The purpose of the dataset is to address data and knowledge gaps in temperature fluctuations in intertidal regions that are influenced by seasonal factors, daily tidal oscillations, as well as large and small scale atmospheric processes. In this way, the dataset can be used to help advance our understanding of the processes affecting the ecology rocky intertidal zones, and how these processes are influence by broader climate change. Overall Comments: The manuscript describing this dataset is very well written. The authors do a very nice job of explaining why the data are important and relevant and how the field data were collected. I also believe the authors do a good job of interpreting their dataset to the audience. In particular, I like that the data are presented in terms of temperature variability across a range of spatial and temporal scales. For instance, for the temporal dimension, the authors use the data to show the general variation in temperature due to seasonal Earth-Sun relationships (i.e. peak temps occurring slightly after the summer solstice), and the hourly variations arising from the push and pull of the tides. Spatial variability is also highlighted by the data with regional variations due to upwelling and the small-scale effects of local weather. Furthermore, the authors use the data to highlight the broader effects of global atmospheric circulation processes (i.e. ENSO; NAO) on rocky intertidal zone temps. This adds broader impacts of the dataset in that it can help move forward modeling studies to address, for instance, the effects of climate change on these ecosystems. I do not know of any other dataset quite like this one, so I do believe it is new. However, the methods are not necessarily novel. But, that is OK I think. The quality of the dataset are good and the links are accessible. The authors use replicate loggers and run statistics across the datasets with high correlation between sites over the study period. Figures and table look good and are appropriate. Rating: Uniqueness: 1 – Again, I do not know of any other dataset like this one, and I do believe that it "provides data on a variable that is supposed to reflect changes in the Earth system". Usefulness: 1 – I believe that this data could be used alone to compare to future trends to assess changes in the Earth system, and can also be combined with other datasets (such as those from a coastal zone observatory) to develop numerical models to predict such

changes and assess the forcing mechanisms behind that change. Completeness: 1 – data are complete and do not appear to be split across multiple manuscripts.

RESPONSE: We appreciate to receive such a positive evaluation of our work. Thank you for capturing its value so well.

—

COMMENT FROM REVIEWER: Minor (technical) considerations: L48-49: The wording in this sentence is a bit awkward. Consider rephrasing?

RESPONSE: We have rephrased that sentence as: "In addition, not only average temperature is ecologically important, but its temporal variability as well".

—

COMMENT FROM REVIEWER: L97: "which is the data set on which this paper is based" I'm not sure you need to say this here? Maybe delete this statement for better flow?

RESPONSE: We deem important to state explicitly that such data are the data specifically reported and discussed in our paper. To improve the quality of that expression, we have rephrased it as: "...which is the data set discussed in this paper, being publicly available from the figshare online repository (Scrosati and Ellrich, 2020a)".

—

COMMENT FROM REVIEWER: L117 – 121: consider adding numbers to the two exceptions. It was a little hard to dig these out of the text and I think adding a number between each will help pull it out and make it clearer for the reader. For instance, "...with just two exceptions: 1) the period between 20 March and 12 April 2017 for L1 because of logger removal by drift sea ice coming from the Gulf of S. Lawrence, and 2) the period between 30 September. ..."

RESPONSE: We have added parentheses (as shown in the revised manuscript) in

convenient places in that sentence to improve the description of those two exceptions.
—

We thank this reviewer for the constructive evaluation of our work, which we are stating in the revised Acknowledgements section.

---

## Author Comment (AC4) · 20 Sep 2020

We have submitted our response to reviewers 1 and 2, but we do not see any button to submit the revised manuscript, which obviously the Editor will need to evaluate our revisions properly. Therefore, we are hereby attaching our revised manuscript (20 September 2020).

Please also note the supplement to this comment:
https://essd.copernicus.org/preprints/essd-2020-161/essd-2020-161-AC4-supplement.pdf

[Figure]

[Figure]

**Supplement:**

[revised manuscript text omitted]